# Iron Oxide Nanoparticles for Visualization of Prostate Cancer in MRI

**DOI:** 10.3390/cancers14122909

**Published:** 2022-06-13

**Authors:** Avan Kader, Jan O. Kaufmann, Dilyana B. Mangarova, Jana Moeckel, Julia Brangsch, Lisa C. Adams, Jing Zhao, Carolin Reimann, Jessica Saatz, Heike Traub, Rebecca Buchholz, Uwe Karst, Bernd Hamm, Marcus R. Makowski

**Affiliations:** 1Charité—Universitätsmedizin Berlin, Corporate Member of Freie Universität Berlin, Humboldt-Universität zu Berlin, and Berlin Institute of Health, Charitéplatz 1, 10117 Berlin, Germany; jan-ole.kaufmann@charite.de (J.O.K.); dilyana.mangarova@charite.de (D.B.M.); jana.moeckel@charite.de (J.M.); julia.brangsch@charite.de (J.B.); lisa.adams@charite.de (L.C.A.); jing.zhao@charite.de (J.Z.); carolinreimann1990@web.de (C.R.); bernd.hamm@charite.de (B.H.); marcus.makowski@tum.de (M.R.M.); 2Department of Biology, Chemistry and Pharmacy, Institute of Biology, Freie Universität Berlin, Königin-Luise-Str. 1-3, 14195 Berlin, Germany; 3Department of Diagnostic and Interventional Radiology, Technical University of Munich, Ismaninger Str. 22, 81675 Munich, Germany; 4Division 1.5 Protein Analysis, Bundesanstalt für Materialforschung und-Prüfung (BAM), Richard-Willstätter-Str. 11, 12489 Berlin, Germany; 5Department of Chemistry, Humboldt-Universität zu Berlin, Brook-Taylor-Str. 2, 12489 Berlin, Germany; 6Department of Veterinary Medicine, Institute of Veterinary Pathology, Freie Universität Berlin, Robert-von-Ostertag-Str. 15, Building 12, 14163 Berlin, Germany; 7Division 1.1 Inorganic Trace Analysis, Bundesanstalt für Materialforschung und-Prüfung (BAM), Richard-Willstätter-Str. 11, 12489 Berlin, Germany; jessica.saatz@bam.de (J.S.); heike.traub@bam.de (H.T.); 8Institute of Inorganic and Analytical Chemistry, Westfälische Wilhelms-Universität Münster, 48149 Münster, Germany; rebecca.buchholz@uni-muenster.de (R.B.); uk@uni-muenster.de (U.K.); 9School of Biomedical Engineering and Imaging Sciences, King’s College London, St Thomas’ Hospital Westminster Bridge Road, London SE1 7EH, UK

**Keywords:** molecular imaging, prostate cancer, iron oxide nanoparticle

## Abstract

**Simple Summary:**

Magnetic resonance imaging (MRI) is a non-invasive method and can be used to diagnose prostate cancer (PCa). Due to their high biological safety, iron oxide nanoparticles are becoming increasingly important as contrast agents for MRI. Macrophages are able to take up these iron particles, which leads to a loss of signal in T2- and T2*-weighted images during MRI. Macrophages play an important role in the development and progression of prostate cancer. In this article, ferumoxytol is visualized at two different PCa volumes on MRI in a xenograft mouse model. Ferumoxytol is a superparamagnetic iron oxide probe and was used here as a contrast agent. The in vivo data were correlated with histological data. When using ferumoxytol, we found that small tumors took up more ferumoxytol than larger tumor volumes. These results were obtained in vivo as well as ex vivo.

**Abstract:**

Prostate cancer (PCa) is one of the most common cancers in men. For detection and diagnosis of PCa, non-invasive methods, including magnetic resonance imaging (MRI), can reduce the risk potential of surgical intervention. To explore the molecular characteristics of the tumor, we investigated the applicability of ferumoxytol in PCa in a xenograft mouse model in two different tumor volumes, 500 mm^3^ and 1000 mm^3^. Macrophages play a key role in tumor progression, and they are able to internalize iron-oxide particles, such as ferumoxytol. When evaluating T2*-weighted sequences on MRI, a significant decrease of signal intensity between pre- and post-contrast images for each tumor volume (*n* = 14; *p* < 0.001) was measured. We, furthermore, observed a higher signal loss for a tumor volume of 500 mm^3^ than for 1000 mm^3^. These findings were confirmed by histological examinations and laser ablation inductively coupled plasma-mass spectrometry. The 500 mm^3^ tumors had 1.5% iron content (*n* = 14; σ = 1.1), while the 1000 mm^3^ tumors contained only 0.4% iron (*n* = 14; σ = 0.2). In vivo MRI data demonstrated a correlation with the ex vivo data (R^2^ = 0.75). The results of elemental analysis by inductively coupled plasma-mass spectrometry correlated strongly with the MRI data (R^2^ = 0.83) (*n* = 4). Due to its long retention time in the blood, biodegradability, and low toxicity to patients, ferumoxytol has great potential as a contrast agent for visualization PCa.

## 1. Introduction

Prostate cancer (PCa) is a malignant and heterogeneous tumor disease, being one of the most common fatal cancers in men [1]. Biomarker testing is often used as a screening test, whereby the prostate-specific antigen (PSA) value is determined from a blood sample. In addition, determination of the prostate health index (PHI) and 4K score can help distinguish between indolent and progressive prostate tumors [2]. Magnetic resonance examinations provide a non-invasive and non-ionizing method to diagnose PCa [3].

Early and reliable detection of tumors based on non-invasive methods is one of the main goals of current tumor research. Multifunctional magnetic nanoparticles can be used in different therapies but also for imaging methods, including magnetic resonance imaging (MRI) [4]. MRI has a high-spatial resolution, is noninvasive, and can be performed in vivo. Iron-containing contrast agents, such as iron oxide nanoparticles, show high biosafety and can be visualized in T2 and T2*-weighted images [4].

After administration of superparamagnetic iron oxide nanoparticles (SPIONs), SPIONs accumulate in the tissue, decreasing the transverse relaxation time of T2, which leads to increased magnetic susceptibility and signal loss in T2 and T2*-weighted images [5,6,7].

Compared to other SPIONs, Feraheme^®^ (ferumoxytol) shows high stability due to reduced carboxymethyl dextran, uniform particle size, and an improved safety profile. Ferumoxytol exhibits a core size of 3 nm to 12 nm [4,8] and a particle size of 17–30 nm. When these particles enter the blood circulation, they will be internalized by activated macrophages. Macrophages have one of the most important functions of the innate immune response and can be involved in the development of tumors [9]. Due to the microenvironment, macrophages can be pro-inflammatory (M1) or anti-inflammatory (M2). The distinction between M1 and M2 can always be clearly differentiated by the variability in the molecular expression profile. In tumors, tumor-associated macrophages (TAMs) also play a key role. Circulating monocytes mature into macrophages. When these migrate in the tumor microenvironment they can be recruited to TAMs [10]. During tumor progression, they influence various processes, such as angiogenesis, tumor cell proliferation, and metastasis [11]. Several research groups have shown a correlation between the uptake of ferumoxytol by TAMs, therapeutic response, and cancer stage [12,13]. The uptake of ferumoxytol by TAMs could be an important step for molecular imaging. The phagocytic cells store the ferumoxytol nanoparticles in secondary lysosomes [14,15]. The carboxymethyl dextran coating is cleaved by dextranase and can be completely excreted by the kidneys. The iron nucleus is integrated into the body’s iron store and is used for cell metabolism and hemoglobin synthesis [15].

Biocompatible magnetic iron oxide nanoparticles (MNPs) have the potential to improve tumor evaluation. Tse et al. developed an MNP conjugated with J591 [16]. J591 is an antibody against an extracellular epitope of prostate-specific membrane antibody (PSMA). It was shown that tumors in vivo enhanced magnetic resonance contrast of tumors with PSMA targeting MNPs. Conjugated superparamagnetic iron oxide particles to target PSMA showed promising results in in vitro tests [17]. Zhu et al. synthesized a PSMA-targeting polypeptide CQKHHNYLC conjugated with SPIONs [18].

In vivo studies enhanced the MRI signal in PSMA-expressing tumors. Histological examination showed heterogeneous deposition of SPIONs in prostate tumor tissue. Modified SPIONs indicate tumor-specific targeting, but not all tumors express these antibodies or proteins, e.g., PSMA [19].

This study investigated the applicability of ferumoxytol for MRI imaging in a xenograft prostate cancer mouse model. Two tumor sizes, 500 mm^3^ and 1000 mm^3^, were investigated. The imaging results were analyzed and compared with histological and biochemical measurements.

## 2. Materials and Methods

### 2.1. Cell Culture

PC3 cells were purchased from ATCC^®^ CRL-1435™ (Manassas, VA, USA) and grown in Roswell Park Memorial Institute (RPMI) 1640 medium (Gibco™, Thermo Fischer Scientific, Waltham, MA, USA) and supplemented with 10% fetal calf serum (FCS) (Gibco™, Thermo Fischer Scientific, Waltham, MA, USA). The genetic information of the cell line are available on the website from ATCC^®^. Cells were cultured in 150 cm^3^ tissue culture flasks until they were about 80% confluent. Then, the cells were washed with phosphate-buffered saline (PBS) (Gibco™, Thermo Fischer Scientific, Waltham, MA, USA), trypsinized and resuspended in 1 mL RPMI medium. Cells were grown at 37 °C and 5% CO_2_. To count the cells, 0.4% trypan blue solution (Gibco™, Thermo Fischer Scientific, Waltham, MA, USA) was used.

### 2.2. Xenograft Mouse Model and In Vivo Study Design

Experiments were performed in accordance with the local guidelines and regulations for the implementation of the Animal Welfare Act and the regulations of the Federation of Laboratory Animal Science Associations (FELASA). Animal experiments were approved by the supervisory authority of the Berlin State Office for Health and Social Affairs (LAGeSo) (G0094/19). Eight-week-old male SCID mice (CB17/Icr-Prkdcscid/IcrIcoCrl) were obtained from Charles River Laboratories (Sulzfeld, Germany) (*n* = 28). The animals were randomly divided into two separate groups (*n* = 14).

For anesthesia, medetomidine (500 µg/kg), midazolam (5 mg/kg), and fentanyl (50 µg/kg) were injected intraperitoneally into the mice for anesthesia. A cell suspension containing 2 × 10^6^ PC3 cells was injected subcutaneously into the right scapula area. Anesthesia was subsequently antagonized with atipamezole (750 µg/kg), flumazenil (0.5 mg/kg), and naloxone (1200 µg/kg).

In vivo MR imaging was performed when the tumor size reached 500 mm^3^ (*n* = 14) or 1000 mm^3^ (*n* = 14), respectively. Tumor size was measured with a caliper. The in vivo imaging was obtained on two consecutive days. On day one, first a native MRI acquisition took place followed by the administration of the contrast agent, ferumoxytol, via the tail vein. Subsequently, the anesthesia was antagonized. A second MRI was performed for the detection and imaging of the accumulation of ferumoxytol 24 h later on day two. After MRI, mice were euthanized, and tumor tissues were collected for ex vivo studies. Figure 1 demonstrates the study design.

### 2.3. In Vivo MRI

In vivo MR imaging was performed using a 3.0 Tesla MR scanner (MAGNETOM Lumina, Siemens, Erlangen, Germany) and a 4-channel receive coil array for mouse body applications (Mouse scapula Array, P-H04LE-030, version1, Rapid Biomedical GmbH, Rimpar, Germany). Following intraperitoneal anesthesia, we positioned the mice on the MRI patient table in the prone position. For administration of the contrast agent during MR imaging, venous access was established via the tail vein. Body temperature (37 °C) was monitored using an MR-compatible heating system (model 1025, SA Instruments Inc., Stony Brook, NY, USA) to prevent rapid cooling.

### 2.4. Ferumoxytol as a Contrast Agent for MRI

Feraheme^®^ (Ferumoxytol) (AMAG Pharmaceuticals, Waltham, MA, USA) is a superparamagnetic iron-oxide particle preparation used in adult patients for the treatment of iron deficiency anemia. The preparation can also be used off-label as a contrast agent in MRI examinations [20]. Ferumoxytol was commercially purchased and used for this study. In this study, it was used as a contrast agent for MRI, leading to a large decrease in T1, T2, and T2* relaxation times [15,21]. Feraheme^®^ has a prolonged blood pool phase with a plasma half-life of 14–21 h. The iron particles are taken up intracellularly with a time delay, which allows MR imaging after 24 h. Mice were administered a clinical dose of 4 mg/kg ferumoxytol via the tail vein.

### 2.5. Ferumoxytol Imaging Using T2* Weighted Sequences

MR imaging was performed with a 3.0 Tesla MR scanner. Following anesthesia, the mice were positioned in the prone position and examined using a 4-channel receive-coil array. For the localization of the tumor in low resolution, a three-dimensional localizer scan was used, which was performed in sagittal, coronal, and transverse orientation with the following parameters: field-of-view (FOV) = 280 × 280 mm, matrix = 320, slice thickness = 1.5 mm, repetition time (TR) = 11.0 ms, echo time (TE) = 5.39 ms, flip angle = 20°, and number of slices = 10. T1-weighted anatomical images were acquired using the following parameters: FOV = 70 mm, matrix = 131, slice thickness = 0.4 mm, TR = 833.8 ms, TE = 6.34 ms, flip angle = 30°, and number of slices = 30. To visualize the iron-oxide-based contrast agent, a T2*-weighted sequence with the following parameters: FOV = 150 mm, matrix = 201, slice thickness = 1.2 mm, TR = 3200.0 ms, TE = 77.0 ms, flip angle = 140°, and number of slices = 25.

Native MRI acquisition took place on day one and contrast-enhanced MRI on day two. After the first MRI pre-contrast session, the contrast agent was administered via the tail vein. Subsequently, the anesthesia was antagonized, and a second MRI was performed 24 h later.

### 2.6. MRI Measurements

MR images were evaluated using Visage 7.1 (Version7.1, Visage Imaging, Berlin, Germany). The T2*-weighted images were analyzed before and 24 h after the administration of the contrast agent (signal loss = *Sl*). For relative reduction (*RR*) assessment, 2D regions of interests (ROIs) were drawn around the respective areas in pre-contrast and post-contrast MR images. The following formula was used to calculate the relative reduction (*RR*):$$RR=\frac{\left(Sl_{precontrast}-Sl{\;}_{postcontrast}\right)}{Sl{\;}_{precontrast}}$$

### 2.7. Histological Analysis

Frozen tumor samples were cut in 9 μm thick serial sections at −20 °C. Sections were then fixed with cold acetone (≥99%, Fisher scientific, Hampton, VA, USA) for 6 min at −20 °C. To visualize the iron ions, a Perls’ Prussian blue stain was performed. Immunofluorescence staining was used to assess the localization and density of macrophages. The tumor tissue was cut in 9 μm thick serial sections at −20 °C on SuperFrost Plus adhesion slides (Thermo Scientific, Waltham, MA, USA). The sections were incubated overnight at 4 °C with a monoclonal CD68 antibody (1:100) (rat anti-mouse CD68, clone FA-11, Bio-Rad, Hercules, CA, USA) diluted in Dako REAL^TM^ Antibody Diluent (Dako, Denmark). Following, slides were washed three times with PBS (pH = 7.4). For macrophage visualization, slides were incubated for one hour with AlexaFluor 568 polyclonal secondary antibody diluted 1:200 (goat versus rat IgG, Thermo Fisher Scientific, Massachusetts, USA), then counterstained, and mounted with Roti^®^-Mount FlourCare (Carl Roth, Karlsruhe, Germany). In conclusion, the sections were analyzed using a Keyence microscope (BZ-X800 Series, Osaka Prefecture, Japan).

### 2.8. Quantification of the Iron and Macrophages in Immunofluorescence

Quantification of staining of iron and immunofluorescence sections was measured using BZ-X800 Analyzer image analysis software (Keyence, Osaka Prefecture, Japan). Three representative areas (two peripheral areas and one central area) were analyzed for each sample. The mean value was calculated in each case. First, the entire region of interest was marked. Then, all stained iron particles or macrophages were identified, and this ratio to the total labeled tumor region was calculated using the labeled pixels.

### 2.9. Laser Ablation-Inductively Coupled Plasma-Mass Spectroscopy (LA-ICP-MS)

Localization of iron in tumor tissue was performed by LA-ICP-MS (*n*=3 per group). The tumor samples were cut into 9 µm cryosections at −20 °C and mounted on SuperFrost Plus slides (Thermo Scientific, Waltham, MA, USA). LA-ICP-MS analysis was performed with a LSX 213 G2+ laser system (CETAC Technologies, Omaha, NE, USA) equipped with a two volume HelEx II cell connected via Tygon tubing to an ICPMS-2030 (Shimadzu, Kyoto, Japan). Line-by-line scanning of the samples was used with a spot size of 30 µm, a scan speed of 90 µm/s, and 800 mL/min. He was used as the transport gas. The analysis was performed in collision gas mode with He as the collision gas and 50 ms integration time for the ^57^Fe isotope. For the quantification of Fe, matrix-matched standards based on gelatin were used. Nine gelatin standards (10% *w*/*w*), including a blank, were spiked with different Fe concentrations, ranging from 7 to 1300 µg/g. Averaged intensities of the scanned lines of the standards showed good linear correlation with the regression coefficient, R^2^ = 0.996, within this concentration range. Limit of detection (LOD) and limit of quantification (LOQ), calculated with the 3σ- and 10σ-criteria, were 3.4 µg/g and 11.4 µg/g Fe, respectively. The quantification and visualization were performed with an in-house developed software (written by Robin Schmid, University of Münster, Münster, Germany).

### 2.10. Inductively Coupled Plasma-Mass Spectrometry (ICP-MS)

ICP-MS was used to determine the total iron (Fe) concentration in the tumor samples. A section of tumor probe was prepared (*n* = 6 per group) and dehydrated under a vacuum atmosphere (vacuum-pumping device, vacuubrand, Wertheim, Germany). To each of the samples, 1 mL of 66% nitric acid was added, followed by incubation at room temperature until the tissue was completely dissolved. Afterwards, deionized water was then added to each sample.

The digested samples were diluted in nitric acid (sub-boiled) 1% and analyzed with an iCAP Qc ICP quadrupole mass spectrometer (Thermo Fisher Scientific, Bremen, Germany) in combination with the autosampler, SC4-DX (ESI Elemental Service & Instruments GmbH, Mainz, Germany), using a 200 µL PFA nebulizer and a cyclone spray chamber. Measurements of the isotopes ^56^Fe, ^57^Fe were performed in KED mode using a nebulizer gas flow rate of 1.08 L/min and a helium flow rate of 5 mL/min as the collision gas. Calibration was carried out in the concentration range of 0.1–50 ng/L with the diluted iron ICP standard CertiPUR (Merck KGaA, Darmstadt, Germany); rhodium was used as the internal standard.

### 2.11. Western Blot

For the western blot, tumor pieces (*n* = 3 per group) were first homogenized in RIPA buffer (50 mM Tris·HCl (Carl Roth GmbH, Karlsruhe, Germany), 150 mM NaCl (Carl Roth GmbH, Karlsruhe, Germany), 0.1% SDS (Carl Roth GmbH, Karlsruhe, Germany), 1% sodium deoxycholate (Carl Roth GmbH, Karlsruhe, Germany), 1% Triton X-100 (Merck, Darmstadt, Germany), and Protease Inhibitor I and Protease Inhibitor II (Thermo Fisher Scientific, Waltham, MA, USA). The samples were shaken for 2 h at 4 °C followed by centrifugation at 12.000 RPM for 20 min at 4 °C. The samples were filtered using syringe filters (1 µm, 0.45 µm and 0.1 µm). The concentration was determined using the BC assay protocol (Pierce™ BCA Protein Assay Kit, Thermo Fisher Scientific, Waltham, MA, USA). The same protein amount (50 µg) was loaded into the wells of the gel under unreduced conditions (SERVAGel™ TG 8% PRiME™, Heidelberg, Germany) and separated in the running gel system (SERVA™ Heidelberg, Germany) at a voltage of 70 V for 60 min followed by 60 min at 160 V in running buffer (250 mM TrisBase (Carl Roth GmbH, Karlsruhe, Germany), 1.92 M glycine (Carl Roth GmbH, Karlsruhe, Germany), and 1% SDS (Carl Roth GmbH, Karlsruhe, Germany)). Subsequently, the proteins were transferred from sodium dodecyl sulphate (SDS) gel to a nitrocellulose membrane (Trans-Blot^®^ Turbo™ RTA Mini PVDF Transfer Kit, Bio-Rad Laboratories, Hercules, CA, USA). The blot system, Trans-Blot^®^ Turbo™ (Bio-Rad, Laboratories, Hercules, CA, USA), was used. A 5% skimmed milk powder (Carl Roth GmbH, Karlsruhe, Germany) in 0.05% PBS-Tween20 (PBS-T) (Carl Roth GmbH, Karlsruhe, Germany) solution was used to block non-specific antibody binding. Incubation was performed at room temperature for 1 h. Blots were incubated with an antibody marker for macrophages, CD68 (Bio-rad MCA1957, Hercules, CA, USA), diluted 1:500 in 5% milk solution overnight at 4 °C. After washing the membrane three times with PBS-T, the blots were incubated with HRP-coupled Mouse IgGκlight chain binding protein diluted 1:5000 in PBS-T for 1 h. The band was detected using the membrane substrate (SeramunBlau^®^ prec, Seramun Diagnostica GmbH, Heidesee, Germany). GAPDH (Invitrogen, Carlsbad, CA, USA) was used for charge control.

The intensity of the bands was measured with the software Image J (version: 1.53k).

### 2.12. Statistical Analysis

From all data, a mean inset was calculated and presented. The significance was compared by unpaired and bilateral t-test analysis and was indicated at *p* < 0.05. Statistics were performed using Microsoft Excel (version: 16.57; Microsoft, Washington, DC, USA).

## 3. Results

In this study, an iron-containing contrast agent was applied to mice with PCa to visualize macrophages in vivo on MRI. Two tumor sizes, 500 mm^3^ and 1000 mm^3^, were studied and compared. Figure 1 shows the study design.

All animals developed a tumor (*n* = 28). Although the amounts of cells and the time frame of the study were standardized, the animals developed the target tumor volume at different time points. To determine tumor size, the tumors were measured daily with a caliper or palpatory and documented. The final tumor size of 500 mm^3^ was reached between 30–58 days after surgery. The tumor size of 1000 mm^3^ was reached after 44–61 days after cell implantation.

### 3.1. Characterization in T2*-Weigthed MR Imaging Using Superparamagnetic Iron-Oxide Particle

After a pre-scanning of the mouse in MRI, intravenous administration of 4 mg/kg ferumoxytol was performed via the tail vein. After 24 h, a second MRI examination was performed. The comparison between the pre-contrast and post-contrast images (Figure 2A,B) showed a signal loss after administration of ferumoxytol in the tumor tissue. Mice with a 500 mm^3^ tumor showed a higher signal loss than mice with a 1000 mm^3^ tumor after administration of ferumoxytol. In the group with a tumor volume of 500 mm^3^, the pre-contrast imaging demonstrated a signal loss (Sl) of 721. After the administration of ferumoxytol, a Sl of 139 (*p* < 0.001) was shown (Figure 2C). In 1000 mm^3^ tumors, the Sl before ferumoxytol administration was 521 compared to 204 after 24 h (*p* < 0.001) (Figure 2C). The RR was 0.8 in 500 mm^3^ tumors (*n* = 14) compared to a RR of 0.6 in 1000 mm^3^ tumors (*n* = 14).

### 3.2. Ex Vivo Analysis

In both tumor sizes, iron particles were detected in the tissue using Perls’ Prussian blue stain (Figure 3A). The iron particles are shown in blue. To determine the iron content, three different areas were selected for each slide and the percentages of the presence of iron were determined using the analyzer. The analysis revealed a different amount of iron between the two tumor sizes. The 500 mm^3^ tumors had an average of 1.5% iron (σ = 1.1, *n* = 14), and the 1000 mm^3^ tumors showed only 0.4% iron (σ = 0.2, *n* = 14) (Figure 3D). The iron particles were distributed differently in the tumor tissues.

In addition, immunofluorescence staining by using antibodies to CD68 was performed (Figure 3B) to confirm the results. This showed that in 500 mm^3^ tumors the mean value was 9.6% CD68 (σ = 1.1, *n* = 3) and in 1000 mm^3^ tumors 5.2% (σ = 0.9; *n* = 3). Additional Western blot analysis was conducted in which the previous results were confirmed. A stronger expression of CD68 was detected in the 500 mm^3^ tumors, mean value of 703.7 (*n* = 3), than in the 1000 mm^3^ tumors, mean value of 304.8 (*n* = 3) (Figure 3E). The intensities of each band can be seen in the Appendix A. GAPDH was included as a control.

The in vivo MRI data (T2-weighted MR sequences) were correlated with the ex vivo data (percent Fe by Perls’ Prussian staining). A correlation was found (y = 0.025x + 0.76; R^2^ = 0.74) (Figure 4A).

### 3.3. Elemental Analysis of Tumor Tissue with Specific Regard to Fe

LA-ICP-MS analysis was performed to localize Fe in the tumor tissue. For each tumor size, *n* = 3 animals were visualized. The LA-ICP-MS data showed good colocalization of the iron-oxide nanoparticles with the histological data, as shown in Figure 3C. Iron could be detected in the intra-tumoral space as well as in the peripheral region of the tumor. There is an overlap between the immunofluorescence staining of CD68 and the LA-ICP-MS measurement for Fe (Figure 3A,B).

To accurately determine the concentration of Fe in the tissue, a quantitative ICP-MS analysis was conducted. The determined iron content in the tumor tissue was correlated with the MRI RR data, which showed a correlation (y = −1.13x + 0.93; R^2^ = 0.83) (Figure 4B). For each group, *n* = 6 measurements were performed.

## 4. Discussion

This study investigated the feasibility of using a clinically applicable iron-based probe for molecular MRI as a signal-reducing substance (off-label) to image and characterize a PC3 tumor in vivo in a SCID mouse model. Two tumor volumes were compared, 500 mm^3^ and 1000 mm^3^. The results show that more iron particles were assimilated in the smaller tumor volume than in the larger ones. Regardless of tumor volume, both tumor volumes were found to take up iron. A clear differentiation between healthy and tumorous tissues was possible. The results could be confirmed by ex vivo examination methods.

In our study, we note a heterogeneous distribution of the iron-oxide particles in the tumor during MRI, which means that iron-oxide particles were not specifically found only in the periphery or in a specific area of the tumor. The histological results confirm this assumption. We could detect an overlap of the CD68 positive areas in the immunofluorescence with the positive Fe areas in the LA-ICP-MS measurement. Thus, it can be said that macrophages are able to take up ferumoxytol. We have already been able to show this in other diseases [21,22,23]. Some publications indicate that tumor-associated macrophages (TAMs) are able to take up administered nanoparticles [24,25]. TAMs account for about 50% of the total tumor mass [26]. They promote tumor growth by suppressing immunocompetent cells, inducing neovascularization, and supporting cancer stem cells [27]. TAMs can invade and distribute in the tumor mass so that they may be instrumental in diagnosis, treatment, and therapy. Daldrup-Link et al. demonstrated that ferumoxytol can be used to detect TAMs in MRI in a breast carcinoma mouse model [25].

In this study, a lower uptake of iron-oxide particles in the group with the tumor volume of 1000 mm^3^ was observed. A possible explanation for this is provided by a study by Franklin et al. [28]. They studied macrophages during growth in breast tumors. Among other things, a distinction between TAMs and breast tissue macrophages (MTMs) was investigated using different methods. This showed a correlation between increase in tumor volume with decrease in MTMs [28,29]. Further studies could prove that there is a correlation between the aggressiveness of the prostate tumor and an increased occurrence of TAMs [11,30]. The exact role of TAMs in different types of cancers has not been fully clarified. The role of TAMs in PCa progression is multifactorial, including involved in tumor invasion, increase tumor angiogenesis, tumor proliferation, tumor metastasis, and immune suppression [31,32]. Thus, our results suggest that although TAMs are able to initialize foreign particles, other functions take precedence in increasing tumor volume. Further research is needed to fully understand the role of TAMs in PCa progression and to determine which functions TAMs prioritize at which stage of tumor progression.

Another reason for the lower uptake of iron particles in tumors with a larger volume can be due to necrosis inside a tumor. With the progression of tumor growth, necrosis plays an important role. An important aspect is tumor necrosis factor-α (TNF-α), which influences tumorigenesis and tumor progression [33,34]. High TNF and interleukin-6 (IL-6) levels, which are known to be responsible for the proliferation and metastatic potential of tumor cells, indicate a negative prognosis for the patient [35]. Maolake et al. were able to demonstrate a TNF-α loop in PC3 cells in their study [36]. A high TNF-α concentration inhibited PC3 proliferation, a low TNF-α concentration caused an upregulation of C-C chemokine receptor, which is significantly associated with lymph node metastasis [37,38]. Our study did not investigate these factors, which is a limitation. Further studies are needed to fully investigate the tumor microenvironment after ferumoxytol administration.

Magnetic iron-oxide particles can be used as off-label MRI contrast agents in the clinical setting. In particular, ferumoxytol is currently used as an MRI contrast agent because it has all the positive properties of iron-oxide nanoparticles, including long blood-retention time, biodegradability, and low toxicity [39]. Additionally, the surface coating of iron-oxide nanoparticles can be modified to achieve specific binding. Thus, conjugation of specific tumor binders to iron-oxide nanoparticles can yield to targeted tumor contrast agents. In addition to magnetic and crystalline properties, the essential properties of iron-oxide nanoparticles must also be considered, such as size, surface charge, and lipophilicity [40]. Iron is a physiologically ubiquitous element in the mammalian body and is more easily metabolized than the conventionally used gadolinium-based contrast agents [41].

Ferumoxytol is mainly used as a replacement therapy in the treatment of anemia. In addition, it can also be used in MRI. In a study by Zanganeh et al., the therapeutic effect of the iron-oxide nanoparticles on the growth of early breast cancer and lung cancer metastases in liver and lung was also investigated [24]. The in vivo experiments showed inhibition of subcutaneous adenocarcinoma growth in mice by administration of ferumoxytol. Intravenous administration of ferumoxytol prevented development of liver metastases. An increased presence of M1 macrophages in tumor tissue was determined [24]. The study shows that ferumoxytol can be used in a variety of medical applications.

The tumor microenvironment is essential for progression and metastasis [24,42,43]. Elemental iron plays an important role in this process. In cancers, iron supply, export, and storage are usually impaired [42]. Targeted iron metabolization could be an innovative approach to treat cancer [42].

Ferumoxytol is not only used in imaging techniques for tumor diseases but may also have applications in cardiovascular diseases [21,22,44] and the central nervous system [15]. Different studies have investigated the molecular properties of cardiovascular disease with a combined approach using iron-oxide nanoparticles and a gadolinium-based contrast agent [21,22,44]. A limitation that needs to be considered regarding ferumoxytol is that it is not prostate tumor specific or even tumor specific. A possible solution would be a conjugation with a disease- or tumor-specific antibody [45]. This would require determining the appropriate targeting ligands for PCa, possible examples being the PSA antigen, alpha-methylacyl-CoA racemase [46], or prostate stem cell antigen [47].

In this context, our study shows that MRI visualization with ferumoxytol is possible in PCa and that there is a heterogeneous distribution in the tumor. This method could be used in combination with other diagnostic methods for non-invasive assessment of the molecular nature of PCa.

## 5. Limitations

The study was performed in a xenograft model. An orthotopic mouse model would allow the tumor in the target tissue and thus in a natural microenvironment to be studied. Ferumoxytol is not prostate cancer specific and can also be used as an MRI-contrast agent for other diseases. In addition, this study only investigated the feasibility of ferumoxytol in PCa, and a full investigation of the tumor microenvironment after administration of the contrast agent would have to follow.

## 6. Conclusions

Our study demonstrates a visualization with ferumoxytol, an iron-oxide nanoparticle probe, is feasible for prostate cancer. The study shows that macrophages in smaller tumors take up more iron than in larger tumors. This non-invasive method could help to detect tumors and to identify molecular characteristics.

## Figures and Tables

**Figure 1 cancers-14-02909-f001:**
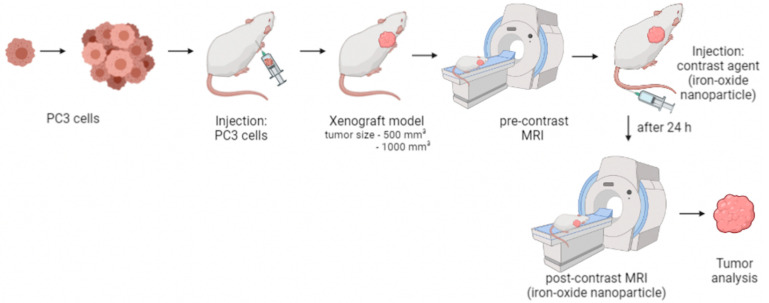
Study design. After reaching the desired tumor size, a pre-contrast image of the tumor was taken. MRI examination was followed by iron-oxide particle injection (ferumoxytol) into the tail vein of the mouse. After 24 h, a post-contrast MRI scan was taken. After in vivo imaging, the tumor was removed and examined.

**Figure 2 cancers-14-02909-f002:**
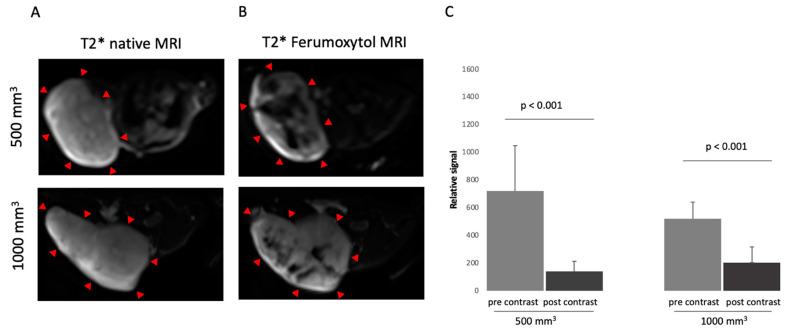
MRI images of prostate cancer in xenograft model of two different tumor sizes, 500 mm^3^ and 1000 mm^3^. (**A**) shows a representative native MRI image of a T2*-weighted sequence from the mouse that developed a tumor with a volume of 500 mm^3^ (top) and 1000 mm^3^ (bottom) in the scapula area of the mouse. Red arrows are pointing at the tumor. (**B**) shows a post-contrast T2*-weighted sequence with ferumoxytol from a tumor-bearing mouse after 24 h. Top: 500 mm^3^. Bottom: 1000 mm^3^. Red arrows show the tumor. (**C**) shows the analysis of MRI images (T2*-weighted sequence) before and after contrast agent administration (ferumoxytol) in two different tumor volumes (500 mm^3^ and 1000 mm^3^). A total of 14 animals per group were studied (*n* = 14).

**Figure 3 cancers-14-02909-f003:**
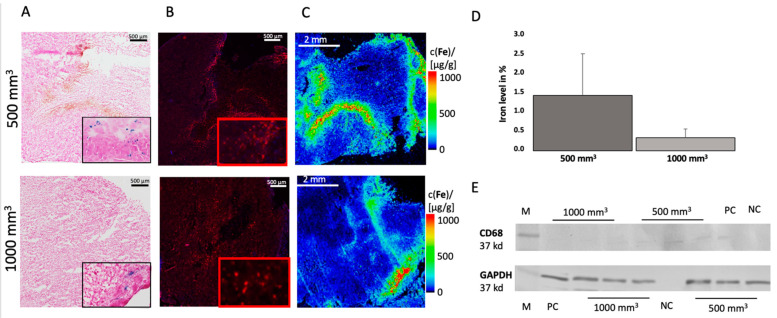
Ex vivo analysis of iron and CD68 in 500 mm^3^ and 1000 mm^3^ tumors. Top: 500 mm^3^. Bottom: 1000 mm^3^. (**A**) shows a Perls´ Prussian blue stain from 500 mm^3^ tumor (top) and 1000 mm^3^ (bottom). Blue colored areas indicate iron particle. (**B**) shows immunofluorescence staining of CD68. Counterstaining was performed with DAPI. (**C**) LA-ICP MS was performed to localize iron particles. (**D**) The percentage of iron in the histological Perls’ Prussian stain was determined. Three areas per slide were calculated and graphically displayed. Per group, *n* = 14 animals were analyzed. (**E**) A Western blot was performed for *n* = 3 tumors per group to detect the expression of CD68, 500 mm^3^ mean value of 703.7 and 1000 mm^3^ mean value of 304.8. A CD38 antibody was used, and GAPDAH was included as a control. M = marker, NC = negative control, PC = positive control. All the whole western blot figures can be found in the Appendix A.

**Figure 4 cancers-14-02909-f004:**
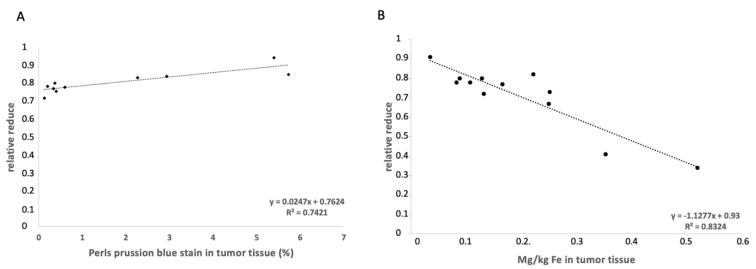
Correlation of the MRI data and the histological and elemental analysis. (**A**) The correlation between the Perls’ Prussian blue staining percentage and the relative enhancements show a correlation (y = 0.025x + 0.76; R^2^ = 0.74). (**B**) shows a correlation between the relative enhancement and the ICP-MS analyses for Fe. A correlation is shown (y = −1.13x + 0.93; R^2^ = 0.83).

## Data Availability

The data presented in this study are available on request from the corresponding author.

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
