# Peer review of "Iron Oxide Nanoparticles for Visualization of Prostate Cancer in MRI"

_cancers, 2022, doi:10.3390/cancers14122909_

Round 1

Reviewer 1 Report

In this MS, the authors used ferumoxytol to perform MRI imaging on two prostate cancer xenograft mouse models with different volumes. The experiments showed that the in vivo angiographic data were consistent with the histological data. When using ferumoxytol, the authors found that small tumors took up more ferumoxytol than large tumor volumes, and the results were validated in vitro and in vivo. Therefore, the authors believe that ferumoxytol has great potential as a contrast agent for prostate cancer, especially for the early diagnosis of prostate cancer. The above is the author's important statement of this research work.

However, the most important experimental data of the article do not correspond to the conclusions, and there are also writing errors, which makes this research really not a good work. The problems that exist are as follows:

  1. Figures 2A & 2B Why is there basically no difference in the size of the cross-sectional volume between a tumor volume of 500 mm3 and a cross-sectional volume of 1000 mm3?
  2. For the two groups of mice with tumor volume of 500 mm3 and 1000 mm3, why did the tumor tissue become brighter after the tail vein contrast agent?
  3. There is an error in the original text: After the administrationof ferumoxytol a Sl of 139 (p<0.00 1) was shown (Figure 3A). In 1000 mm3 tumors, the Sl before ferumoxytol administration was 521 compared to 204 after 24 h (p< 0.001) (Figure 3A) Both figures are in 2C but not in 3A.
  4. In Fig. 3E, the positive control of CD68 basically can not see the bands, and it can not be seen from the bands that the CD68 expression level of 500 mm3 is higher than that of 1000 mm3.
  5. R2 in the two correlation analyses in Fig. 4A & B are both less than 0.9, which cannot well explain the correlation.

In addition to the controversial experimental data in this paper, there are many places that need to be supplemented with experiments and explanations:

  1. The nano-iron oxide used in this study comes from the commercial purchase of FerahemeÒ. It is not the author's development and exploration of new materials. It is of little significance for the visualization of such commercialized nanomaterials.
  2. For the purpose of this paper, this research work does not give any data on the morphology characterization of ferumoxytol.
  3. At the end of the abstract, the research work shows that ferumoxytol can be used for the diagnosis of early prostate cancer. But for the mouse model made in this paper, the tumor volume of the two groups of mice is 500 mm3 and 1000 mm3 respectively, such a large volume of tumor tissue cannot be said to be the basis for early contrast imaging and diagnosis.
  4. The angiographic imaging data in Figures 2A & 2B in this paper only selects the time point of 24h. Generally speaking, we recommend taking continuous and closer time points, such as 0h, 2h, 4h, 6h, 8h, etc. to observe the time point. Contrast properties of ferumoxytol.

Reviewer 2 Report

The article “Iron Oxide Nanoparticles for Visualization of Prostate Cancer in MRI” investigates the role of iron oxide-based MRI contrast agent in prostate cancer using a xenograft mouse model.

The paper is good written and the methods as well as the procedures conducted in the study are clearly and detailly presented.   

This research in its current form, however, lacks utility for clinical application, as the work does not provide a reasonable relationship between tumor volume, macrophage number, and ferumoxytol uptake.

The tumor volume does not reflect the histological aggressiveness of the tumor, so we cannot draw a clear conclusion from the results of this study on the iron concentration in tumor tissue. 

Is there a histological reason why macrophages in smaller tumors take up more iron than in larger tumors?

Was there a correlation between iron uptake and the Gleason score of the tumor?

Reviewer 3 Report

The manuscript describes a MRI-based strategy to detect prostate cancer in a xenograft animal model. The authors use low toxic, biocompatible, and commercially available iron oxide nanoparticles to visualize signal contrast in prostate lesions of two different sizes. Results showed the iron oxide nanoparticles led to significantly reduced MRI signal in both tumor volumes and the small tumors had higher signal loss than the large counterpart. The real-time in vivo imaging results correlate with ex vivo histological analysis, laser ablation inductively coupled plasma-mass spectrometry, and elemental analysis. The authors discussed possible mechanisms for iron oxide uptake. However, the authors did not mention and discuss prior research with iron oxide nanoparticles in prostate cancer imaging, while there are many reports, even some with active targeting strategy with surface modified iron oxide nanoparticles, such as Nanomedicine. 2015, 10, 375-86; Int J Mol Sci. 2015, 16, 9573–9587; BMC Cancer. 2012, 12, 284. In addition, the authors need to address the following issues:

  1. How do the authors determine the tumor size to be exact 500 mm3 or 1000 mm3? Is there supposed to be a deviation in the tumor volume even with a caliper?
  2. Please explain the signal loss in Figure 2. “Signal loss of 721, 139” is a relative number or calculated by a proposed method? Why do the baseline conditions before nanoparticle injection also show a signal loss? The legend “The data are significant” does not make any sense since there is no comparison.
  3. In the abstract section, no need to mention the two equations.
  4. Please reword the sentence “Using the ferumoxytol uptake by TAMs for molecular imaging of tumors could be an important step for diagnostic radiology”; “a scan speed of 90 µm/s and 800 mL/min He as transport gas”; tryptan blue should be trypan blue.
  5. Low volume tumors showed relatively higher iron oxide nanoparticle uptake. Is it possible that large tumors have a lower EPR effect due to tumor necrosis?
  6. Please pay attention to inconsistency or typos, such as FerahemeÒ vs. Feraheme, United States)), 12.000 rpm.
  7. Since iron oxide is heterogeneously distributed in the tumor region, the authors chose three different areas in each slide to determine the iron oxide content. How can the data reliably suggest that the relative content of iron oxide in small tumors is higher? Furthermore, the error bar in Figure 3D for the small tumors is quite large. Please indicate the sample size for σ = 1.1 and σ = 0.2.
  8. In the discussion section, the authors focus too much on ferumoxytol. It’s unnecessary since it’s agency approved, and readily available for pre-clinical use. The authors should focus on the limitations of the current study.
  9. Please address the reference format, such as 3, 18, 22, and 33.

Round 2

Reviewer 2 Report

Dear Authors,

Thank you for revising the article and for citing some important studies emphasizing factors that may be responsible for why tumors with a larger volume have a lower uptake of iron particles.

Author Response

We thank the Reviewer.

Reviewer 3 Report

The authors appropriately addressed the comments and significantly improved the manuscript. However, I have the following comments:

1. I would expect a deviation of the tumor size in each group with n = 14. In addition, not sure whether there is any difference when the tumor size is measured by a caliper and MRI in each group.  

2. Please provide background information in the introduction section regarding prior research using similar iron oxide nanoparticles strategies for tumor imaging. 

3. Please check the reference format. Some with full journal names. Others are abbreviated. 

4. The authors stated that the axis title in Figure 2 was changed from the axis title from “Signal loss” to “Relative signal", but it looks like no change has been made in the revised version.
